# The Mechanical, Thermal, and Chemical Properties of PLA-Mg Filaments Produced via a Colloidal Route for Fused-Filament Fabrication

**DOI:** 10.3390/polym14245414

**Published:** 2022-12-10

**Authors:** Jaime Orellana-Barrasa, Ana Ferrández-Montero, Aldo. R. Boccaccini, Begoña Ferrari, José Ygnacio Pastor

**Affiliations:** 1Centro de Investigación en Materiales Estructurales (CIME), Universidad Politécnica de Madrid, 28040 Madrid, Spain; 2Instituto de Cerámica y Vidrio (CSIC), Campus de Cantoblanco, 28049 Madrid, Spain; 3Institute of Biomaterials, Department of Materials Science and Engineering, University of Erlangen-Nuremberg, Cauerstrasse 6, 91058 Erlangen, Germany

**Keywords:** PLA-Mg, FTIR, DSC, crystallinity, tensile test

## Abstract

The effect of Mg particles on the thermal, chemical, physical, and primarily mechanical properties of 3D-printed PLA/Mg composites is studied in this paper. Recently, new colloidal processing has been proposed to introduce Mg particles into the PLA matrix, which ensures good dispersion of the particles and better thermal properties, allowing for thermal processing routes such as extrusion or 3D printing via fused-filament fabrication. The thermal and physical properties are here studied in 1D single-filament-printed PLA/Mg composites with 0 to 10 wt.% of Mg particles by Differential Scanning Calorimetry (DSC); we analyse the PLA chain modifications produced, the crystallinity fraction, and the different crystalline forms of the PLA after thermal processing. Fourier Transform Infrared Spectroscopy (FTIR) is used to confirm the influence of the PLA/Mg colloidal processing after printing. The mechanical properties are measured with a universal tensile test machine on the 1D single-printed filaments via fused-filament fabrication (FFF); the filaments were naturally aged to stable conditions. Filaments with and without a notch are studied to obtain the materials’ tensile strength, elastic modulus, and fracture toughness. Different analytical models to explain the results of the PLA-Mg were studied, in which the minimum values for the interface strength of the PLA-Mg composites were calculated.

## 1. Introduction

PLA-Mg composite materials are promising biodegradable biomaterials for improving the healing process of bone tissues. The repair of damaged bone remains a significant concern. This has led to a large amount of research on topics including bone grafts [1], bone cement [2,3], scaffolds [4,5], hydrogels [6], or cell treatments [7], in an effort to regenerate bone when the damage incurred is considerable, compared with its healing capacity. Adding growth factors or other chemicals, such as calcium phosphates or Mg, can enhance the regenerative properties of biomaterial strategies.

Mg is an essential component of the human body. Since 1935, it has been used as a biomaterial [8]. One of its roles is to support the formation of bone tissue by promoting both the proliferation and differentiation of stem cells into osteoblasts. Mg has osteogenic properties [9,10]; because of this, it can be used to promote the healing or recovery of damaged bone tissue in patients with any bone defects, incluing traumas, tumours, or inflammations, which necessitate the formation of new bone. However, introducing Mg directly into the body can produce issues related to the release and local accumulation of hydrogen, a degradation product of Mg, which is a highly corrodible metal [10,11,12,13].

Moreover, for the optimum healing conditions for bone tissue, it has been found that controlling the degradation rate of the Mg implant is essential, as the implant also provides mechanical support [14]. One option for exploiting the osteo-promotive benefits of Mg while controlling its fast degradation rate is to create a composite in which the Mg particles are dispersed in a bioabsorbable material that slowly releases the Mg [15]. For this purpose, PLA is an excellent material, as it is biocompatible, and its degradation rate inside the human body can be tuned.

PLA is also a common material used in additive manufacturing, specifically in fused-filament fabrication (FFF) processes. In this thermal extrusion process, the PLA-matrix is melted, extruded in specific locations, and then cooled down. This process can also be undertaken with the PLA-Mg composite, and implants or scaffolds can be produced that perfectly match the damaged bone. This strategy, of using PLA as a matrix and then printing structures, has previously been explored by loading the PLA with other organic and inorganic materials such as chitosan [16], bioactive glass [17], graphene [18], carbon nanotubes [19], cellulose [20], and hydroxyapatite [21]. 

Regarding previous studies on PLA-Mg composites, some authors have published results on this composite material [22,23,24,25,26,27,28,29,30,31]. Three significant problems described in the previous studies of Cifuentes et al. [27] were: (1) the low interface strength with the increasing amount of Mg particles, (2) the agglomeration of these particles, and (3) the degradation of the PLA. Later studies by Ferrández-Montero et al. [23,24,25,26] indicated that the interface strength between PLA and Mg could be improved by covering the Mg particles with cationic dispersants, thus avoiding the formation of Mg agglomerates. 

Before studying the PLA-Mg filaments obtained through the colloidal route developed by Ferrández-Montero et al. [23,24], a previous study was conducted, which focused on how FFF—the thermal extrusion process—affects the PLA matrix [32]. To decouple the effects of any 3D-printed structure (filament–filament adhesion, the concentration of stresses on the joints, the printing speed, layer height, raster angle, infill, etc. [33]), single 1D filaments were studied; this constituted a novel approach in the literature, which improved the understanding of the material itself. In that research, it was found that, after being printed, the PLA had lower thermal and mechanical properties, which slowly increased with natural ageing until they stabilised at values more commonly associated with PLA. For this research, we aim to understand the effect of adding Mg particles to PLA by studying single filaments (1D structures); this is an approach that has not previously been studied.

This work reproduces, compares, discusses, supports, and extends the current research on PLA/Mg composites by analysing the effect of Mg particles on the neat PLA matrix whilst considering the crystallinity, the stabilisation of crystallinity structures, the effect of the plasticiser, the stress concentration, and the PLA–Mg interface strength. Various experimental procedures were used to ensure that all of the essential concerns in the literature are addressed regarding the effects of Mg particles on the PLA-Mg composites produced through this novel colloidal route.

The study from Cifuentes et al. [27] involving tensile stresses concluded that the Mg slightly modified the elastic modulus but was entirely below the expected values, indicating (from the Voigt model) a potential low bond strength between the PLA and the Mg. This forms a significant part of the discussion in this article, as there are other variables (such as the crystallinity, the ageing, the thermal degradation, or the method used for measuring the bond strength) which are approached from a comprehensive mechanical characterisation of 1D single filaments of the PLA-Mg composites. This comprehensive characterisation includes the calculation of the fracture toughness for different PLA-Mg composites and fractographies of samples broken under tensile tests; none of them were previously reported in the literature for any PLA-Mg composite. The Voigt model is used too, and a comparison with the rest of the literature on PLA-Mg composites is provided.

In summary, this paper introduces new results on how Mg particles introduced through the colloidal route developed by Ferrández-Montero et al. [23,24,25,26] affect the properties of the PLA matrix through DSC, FTIR, and tensile tests with and without a notch, complemented with fractographies of samples broken under tensile tests. Modelling of the tensile strength and elastic modulus are also provided, in order to further support our conclusions on the PLA-Mg interface. This work supports and extends other research results by offering an in-depth analysis of printed PLA-Mg composites.

## 2. Materials and Methods

As stated above, single PLA-Mg filaments were studied after being printed via FFF. By studying these single filaments, it was possible to decouple the effect of the structure on the properties from those solely related to changes in the material. The following experimental procedures were used, attending to the information they could provide. 

### 2.1. Production of the Material

PLA filaments, 1.75 mm in diameter, were produced by melting and extruding the PLA pellets supplied by Nature Works (Ingeo biopolymer 2003D, a high molecular weight PLA with M_w_ 182.000 g/mol and a 4.25% content in d-lactide [32]). We followed the colloidal route, described in more detail in the literature [28,29], for producing the 1.75 mm PLA-Mg filaments, obtaining the PLA-Mg composite schematized in Figure 1.

The PLA and PLA-Mg filaments were thermally extruded via FFF at 190 °C and 155 °C, respectively, to form 1D filaments, in ambient humidity and at room temperature (23 ± 1 °C). Filaments 300 to 400 microns in diameter were obtained. A PRUSA i3 MK3S with a 0.4 mm nozzle was used. Adding 5 wt.% of PEG made the printability of PLA-Mg filaments possible at 155 °C.

Considering our previous study on PLA [32], at least three months are necessary for the natural ageing of neat 2003D PLA, without Mg particles. Thus, all samples were naturally aged inside zip bags with desiccant and protected from solar light radiation for a minimum of 90 days before testing.

### 2.2. Differential Scanning Calorimetry

Differential scanning calorimetry (DSC) was used to determine the impact of the addition of Mg on PLA’s thermal and physical properties. This technique can reveal exothermal and endothermal processes in the material, which can be correlated with the described responses of PLA in the literature, such as crystallisation of the α or α′ form [27,34]. This polymorphism of PLA is essential for its properties, as it is related to its mechanical properties, barrier properties, thermal properties, and degradability properties, among others [35].

DSC was performed with a Mettler Toledo 822e instrument inside a 40 μL aluminium crucible, at a heating rate of 10 °C/min, from 40 to 210 °C, on 5–10 mg of the PLA and PLA/mg composites (the printed filament) cut down in pieces 2–4 mm length. All the materials were carefully placed inside aluminium crucibles, ensuring that all pieces were in contact with the bottom of the crucible. This was important, as it was found through this research that slight shifts (±1 °C) in the temperature peaks can be produced if the configuration of the filaments inside the crucible changes. The material PLA-7.5Mg was evaluated at different heating rates: 1, 3, 10, and 20 °C/min. For the calibration of the DSC device, the Indium standard was used.

### 2.3. Fourier Transform Infrared Spectroscopy

FTIR is an extremely sensitive technique used to detect any variation in chemical bonds [36], and is well described in the literature [37,38]. These chemical bonds affect the configuration and packing of the macromolecules, thus making it an excellent technique for analysing crystallinity, changes related to -OH terminal groups [39], and the chemical bonds between the PLA, PEI, and Mg.

Fourier transform infrared spectroscopy (FTIR) was conducted with an IRAffinity-1S from SHIMADZU. The samples were analysed from 200 scans in the mid-IR (400 to 4000 cm^−1^) in absorbance, apodisation with the Happ–Genzel function, and a resolution of 1 cm^−1^.

### 2.4. Archimedes Test

Archimedes tests were used to calculate the density and porosity of the material, which are highly related to its mechanical properties. Archimedes tests were performed in ethanol in a Mettler Toledo Balance coupled to an LC-Density device, following the same procedure as in our previous study [32]. It was found that more consistent and accurate results were obtained using ethanol, rather than distilled water, due to the improved wettability between PLA and ethanol, which avoids the formation of tiny bubbles around the filaments.

Theoretical density values for each composition were calculated using the rule of mixtures. The difference between the experimental and theoretical results determined each composition’s porosity. For calculating the theoretical densities, the following values were used: 1.240 g/cm^3^ for the PLA, 1.125 g/cm^3^ for the PEG, and 1.738 g/cm^3^ for the Mg.

### 2.5. Tensile Tests, Notch, and Fractographies

Tensile tests on filaments with and without a notch were performed. Tests on samples without a notch provided information on the elastic modulus and tensile strength. Tests on samples with a notch provided information on the fracture toughness of the materials, which has not been previously reported for any PLA-Mg composite and ultimately controls the tensile strength of PLA.

All samples were tested following the same method for the tensile tests described in our previous study [32], following the UNE-EN ISO 527-1:2019 standard: samples of 20 mm were tested at 1 mm/min with a 1 kN load cell in an INSTRON 5866, glued to cardboard to avoid inducing mechanical damage with the clamps, equipped with rotulas to prevent stresses due to rotations or bends, and aged for a minimum of 90 days at room temperature inside zip bags with desiccant. A scheme of the tensile test is shown in Figure 2a. A NIKON Profilometer with a resolution of 1 μm was used to measure the dimensions of the filaments.

The fracture toughness (K_IC_) was measured for the filaments by making a notch perpendicular to the filament’s longitudinal direction with the help of a sharp blade (coherent with the indications of the European Structural and Integrity Society (ESIS)) and a carved plate of steel, as schematised in Figure 2b. The NIKON profilometer was used to measure the dimensions of the notches. Although each sample was carefully analysed before and after the fracture, the notch depths were approximately half the filament diameter. Tensile tests were performed on filaments with notches under the same conditions described before, and the fracture toughness was calculated using the James and Mills equation [40], based on research group experience.

Details of fracture surfaces on samples broken with and without a notch were taken with an AURIGA FESEM ZEISS, prior to the deposition of a 20-nm-thick carbon coating with a LEIKA carbon metalliser.

It was possible to calculate the theoretical elastic modulus for each material, which was later compared with the measured values of the elastic modulus. This provided information about the interface strength between the PLA matrix and the dispersed Mg particles, which has not been measured in the literature on single PLA-Mg filaments using this method, but instead through FTIR and DSC (chemical interactions), nano-indentation and compression (compression), DMA (dynamic tensile test), or by tensile tests on bulk samples through the Voigt model (tensile test).

The Voigt model—a kind of rule of mixtures—was used in this study to model the elastic modulus (Equation (1)), in which a bond-strength factor was added to the Mg contribution to assess the strength of the bond:E_c_ = BS_%_(E_Mg_V_Mg_) + (E_PLA_(1 − V_Mg_))(1)
where E_c_, E_Mg_, and E_PLa_ are the elastic modulus of the composite, of the Mg particles (45 GPa [41]), and of the neat PLA (3.6 GPa [32]) respectively; V_Mg_ is the volumetric fraction of Mg particles; and BS_%_ is the bond strength (from 0 to 1). Correction factors were applied to the neat PLA elastic modulus, which is discussed through the text, for decoupling different variables, such as the crystallinity or the plastification effect of the PEG, which also affect the elastic modulus. The modelled materials were considered homogeneous and isotropic. The bond strength between the PLA matrix and the Mg particles was calculated by the best fitting to the experimental data, assuming the interfaces between PLA-PEI-Mg to be only one, which is referred to as the PLA-Mg interface from now on.

To model the tensile strength of the PLA-Mg composites, we considered: (1) the crystallinity, (2) the porosity, (3) the PLA-Mg bond strength, as previously calculated, (4) the degradation of the PLA, (5) the stress concentration, (6) the debonding of the Mg particles, (7) the fracture toughness, experimentally measured for each PLA-Mg composition, and (8) the PEG influence, all of this represented in Figure 3. The modelled materials were considered homogeneous and isotropic. Specific details about the tensile strength models are provided throughout the text.

## 3. Results and Discussion

### 3.1. Thermal and Physical Characterisation

Regarding the results obtained from the DSC at 10 °C/min, there are significant differences in the thermal properties when Mg particles are added, as shown in Figure 4. Table 1 summarises different values extracted from the DSC graphs. The glass transition, enthalpic relaxation, crystalline phase, and melting point are indicated in Figure 4.

PLA’s glass transition decreased with the addition of Mg particles, as seen in Figure 4a. The enthalpic relaxation is directly related to the glass transition; consequently, the enthalpic relaxation temperature also decreased. As observed by Pascual-González et al. [22], adding PEG to the PLA matrix decreases the T_g_ of the PLA. Thus, a decrease in the T_g_ was expected in PLA-Mg compositions as the PLA matrix contains PEG. However, the T_g_ kept decreasing with the addition of higher contents of Mg, and the PEG amount remained the same, indicating the existence of another mechanism, different from the PEG, which explains the decrease in T_g_ with the increasing Mg content. The decrease in T_g_ is associated with the enhancement of the polymer chain movement, caused by the decrease in crystallinity in composites with more than 10 wt.% Mg, as was reported previously [23]. This behaviour could also be related to PLA degradation during the 3D printing process, as the existing literature describes a decrease in the molecular weight by depolymerisation with the addition of Mg particles [42], and due to the printing conditions [22]. However, the PLA-Mg composites were printed at 155 °C, a temperature well below its degradation temperature [23], and a PLA degradation process would be detected by a modification in the central melting temperature, but was not detected here.

Another observation that can be made from the DSC values in Table 1 is the higher enthalpic relaxation enthalpy of PLA-Mg composites compared with neat PLA. In our earlier study [32], the enthalpic relaxation enthalpy of neat PLA increased up to 366 days, and no signs of reaching a stable value were observed after one year of natural ageing. As PLA-Mg composites run higher values for the enthalpic relaxation enthalpy after being naturally aged for three months, faster natural ageing of PLA-Mg composites was recorded, compared with neat PLA. Cangialosi et al. [43] described faster ageing due to the addition of silica nanoparticles, explaining this acceleration with the tendency of nanoparticles to occupy the free volume of the polymer, leading to an inefficient packing of polymers [44]. Although Mg particles are not in the scale of nanometres but micrometres (average particle diameter 30 microns), this phenomenon has been reported for PLA/Mg composites [23]. Additionally, the addition of the PEG plasticiser increases the mobility of PLA macromolecules, facilitating both the evolution of the PLA to the equilibrium state and a decrease in free volume, which could also be the reason for the increase in ageing values.

Figure 4b–d show a new crystallisation at lower temperatures with the incorporation of Mg; by visually comparing the DSC graphs in Figure 4, higher crystallisation enthalpies are observed in PLA-Mg composites, compared with neat PLA. That the crystallisation temperature decreases and the crystallisation enthalpies increase with the higher contents of Mg could be explained by the addition of PEG—a plasticiser—to the PLA-Mg composites. As discussed for the glass transition, there is a problem: the amount of PEG remains constant in all the PLA-Mg compositions. Thus, the PEG itself cannot explain the evolution trend of these two properties with increased Mg. In the literature, Mg has been described as a nucleation point for PLA crystals in similar feedstock processed by other techniques, and in this composite feedstock before the 3D printing process [23,24,27]. Mg particles dispersed in the PLA solution produce effective seeding for the PLA precipitation during the feedstock processing. Additionally, Mg particles could produce a higher crystallisation due to the seeding effect during the DSC measurement.

The new crystallisation peaks starting at temperatures around 80 to 90 °C and ending at 115 to 120 °C, shown in Figure 4c, correspond with the beginning and end of the α′ crystallisation [34]. This α′ crystallisation was not observed in neat PLA samples, as shown in Figure 4b, nor in the feedstock before the 3D printing process [23] This indicates the possible stabilisation of the α′ crystal form by a combination of the presence of Mg and the temperature. The DSC indicated as α′ crystallisation has a shoulder to the right side, as shown in Figure 4e. This is consistent with the existence of an α crystallisation in that area, similar to the one observed in neat PLA, shown in Figure 4d. However, due to the overlap of the α and α′ crystallization processes, it is not easy to reach any conclusions regarding the crystallisation with the DSC. Regarding the events indicated in Figure 4f–h, it was observed that a new endothermal process appears at temperatures between 140 and 146 °C, before the melting observed in the neat PLA (Figure 4h). This new process in PLA-Mg is consistent with the melting of previously formed α′-crystals [34], and the melting peak at 151–153 °C corresponds to the melting of the α-crystals observed in the neat PLA.

The α′conversion into α crystals is a thermally driven process in which time is essential. This process involves the dynamics of the PLA macromolecules; thus, faster heating rates allow less time for converting α′ crystals into α crystals. The PLA-7.5Mg was studied at different heating rates on the DSC to further understand the α′-crystal’s stabilisation by the Mg and to try to evaluate its effect on the α crystals. Changing the scanning rates makes it possible to differentiate the events that are happening inside the PLA. Results are shown in Figure 5 and summarised in Table 2.

Regarding the scan at 1 °C/min, in Figure 5, it is observed that the α′ crystallisation and the peak associated with the melting of α′-crystals were confined to the left shoulder on the melting of the α-crystals. This indicates a conversion of α′ into α during the DSC scan, which would be expected if our previous hypothesis for the new peaks related to the α′ in the PLA-Mg composites is correct. Regarding the increasing heating rates (1, 3, 10 and 20 °C/min) in Figure 5, the melting enthalpies of the α′-crystal rise, indicating a higher melting of α′ crystals, and that the α′ conversion into α did not have enough time to be completed during the 3D printing process. 

Looking in detail at the graph at 3 °C/min, in Figure 5, two facts are indicated.

The first fact is the decoupling of the α′ and α crystallisation, Figure 5a; this provides a precise α crystallisation temperature for the PLA-7.5Mg, which could not be defined before for any PLA-Mg composite. Notice that the temperature peaks depend on the scanning rate by comparing the α′ crystallisation temperatures in Figure 4. As a direct comparison with the results in Figure 4 is impossible, a DSC of neat PLA at 3 °C/min was conducted. The α crystallisation temperatures were 100 °C for the PLA-7.5Mg and 115 °C for the neat PLA. With this information, it was concluded that the α crystallisation also happens at lower temperatures in the PLA-7.5Mg compared with the neat PLA, which could not previously be confirmed.

The second process shown in Figure 5b is an exothermal process in the left shoulder of the α melting shoulder that corresponds to the melting of α′. This exothermal peak in this shoulder is related to the energy released during the transformation of α′ into α [45,46,47], which can extend up to temperatures of around 150 °C at heating rates of 2 °C/min [34]. This is consistent with the literature, and proves the existence of an exothermal process happening simultaneously with the endothermal melting of α′.

All the changes in the thermal and physical properties described up to this point—a decrease in T_g_, an increase in the natural ageing rate, a decrease in the α′ crystallisation temperature, and growth of the α′ crystallisation enthalpies—could be explained by those two mechanisms discussed above: (1) the addition of PEG, a plasticiser, and (2) a potential stabilisation of PLA crystal forms by the Mg.

To further understand the PLA and PLA-Mg composites produced through the colloidal process, FTIR spectroscopy was performed to analyse the previously reported interaction between PEI and PLA, the bands related to the α′ and α crystals, and the composite changes after the 3D printing process.

Only slight variations between PLA and PLA-Mg composites were observed, suggesting that the chemical bonds present in PLA and PLA-Mg composites were similar and consistent with the literature [28]. The FTIR results are shown in Figure 6, and details are shown in Figure 7.

Regarding the results for α′ and α crystal structures in the PLA and PLA-Mg composites at room temperature, small shifts were found at peaks related to PLA’s crystallinity. Rodrigues et al. [48] described a shift in the peak at 1748 cm^−1^. Ferrández et al. [23] analysed the peaks at 867 cm^−1^, 1266 cm^−1^, 1382 cm^−1^, and 1748 cm^−1^, related to the polymorphism of PLA, specifically with α and α′ crystals. Our results on these four peaks, represented in Figure 7a–d, were:

Peaks related to α-crystals:At 867 cm^−1^, in Figure 7a, a slight shift from 868 cm^−1^ to 866 cm^−1^ for PLA with Mg particles.At 1748 cm^−1^, in Figure 7d, a split into two peaks at 1754 cm^−1^ and 1746 cm^−1^ for PLA with Mg particles.

Peaks related to α′-crystals:
At 1266 cm^−1^, in Figure 7b, it remains stable.At 1382 cm^−1^, in Figure 7c, it remains stable.

These results are interesting, as it is observed that α-crystals are slightly affected by the addition of Mg during the cooling of the PLA-Mg, but no differences were observed for different Mg contents. The PLA studied here has a high d-lactide percentage (4.25%), and percentages above 2% are expected to produce neglectable crystallinity percentages during the cooling, according to the literature, explaining the minor variation in the FTIR peaks related to α′ and α at room temperature. This low crystallinity at room temperature was also expected in relation to the DSC results reported by Cifuentes [27] during the cooling of PLA-Mg, being consistent with our results here for the PLA-Mg filaments after printing, and further demonstrating that the material obtained after printing via FFF—in the conditions here described—has no crystallinity independently of the Mg content at room temperature after being processed. 

The changes in the PLA matrix can be analysed by observing the peaks associated with the -OH terminal groups of the PLA: 867 cm^−1^ and 1266 cm^−1^ [23], in Figure 7a and Figure 8b. If the PLA is degraded, the number of -OH terminal groups should increase: the shorter the chains, the more terminal groups there are. However, the change in the intensity of those two peaks is negligible for all the compositions studied; without further consideration, this would indicate that the PLA is not degraded with the addition of Mg. Note that PEI was added to improve the PLA-Mg adhesion, as it is a component that forms interactions and, later, a covalent bond with the -OH terminal groups of the PLA, thus decreasing the signal of the -OH terminal groups. 

Additionally, a decrease at 2850 cm^−1^ is observed with the increase in Mg particles in the sample (Figure 7f) and, consequently, with the increase of PEI. This decrease is associated with the interaction of PLA ester groups with -NH on the PEI. This supports the strong interaction between PLA and PEI, previously observed by Ferrández-Montero [23,24]. Regarding the PEI, comparing the samples with Mg with the neat PLA in the FTIR in Figure 7, a peak was observed at 1647 cm^−1^ in samples with Mg, as shown in Figure 7e. This peak is related to the covalent bond between the PLA and PEI [23], as previously reported, which is created during the thermal processes, extrusion, and 3D printing.

The Archimedes test provided information about the density of the PLA and PLA-Mg composites. The results are shown in Figure 8.

The density slightly increases up to 1% with the addition of Mg particles. However, this increase is below the expected theoretical value. This discrepancy is related to the rise of porosity inside the material, later confirmed with SEM images. The closed porosity for each composition was calculated using the difference between the experimental and theoretical density values.

### 3.2. Mechanical Tests and Fractography

Starting with the elastic modulus, due to the higher elastic modulus of the Mg, [the elastic modulus of PLA/Mg composites was expected to increase, from a theoretical point of view. However, the experimental results showed a slight decrease in the elastic modulus with the addition of Mg. A promising parameter for explaining this difference is the bond strength included in the Voigt model (see Equation (1)). To provide a precise value for the PLA-Mg bond strength with the Voigt model, it was necessary to understand and decouple the effect of other variables that affect the elastic modulus, such as the influence of the PEG, the crystallinity percentage, or the porosity.

To remove the effect of the PEG, the evolution trend of the elastic modulus was used when PEG was added to the PLA, as found in the work of Pascual-González et al. [22]. With that trend, an approximated value of 3.5 GPa for the elastic modulus of our PLA-5PEG is the composition of the PLA matrix for all the PLA-Mg compositions.

The effect of the porosity calculated during the Archimedes test was deduced from the calculations by adding a third term to the Voigt model, in which the pore is considered as a third component with an elastic modulus of 0.

The effect of crystallinity on PLA has previously been described in the literature. A trend can be found for the evolution of the elastic modulus with the crystallinity by comparing different works [23,49] (see Figure 9) in which the elastic modulus is normalised by making the extrapolated elastic modulus equal to 1 at 0 % crystallinity for the materials in each work. The trends from those two works indicate that the effect of the crystallinity in the PLA is an increase of approximately 1% on the elastic modulus (with respect to the value at 0% crystallinity) for every 1% increase in the crystallinity.

As observed in the DSC and later confirmed by the FTIR, our samples were almost completely amorphous (crystallinity contents below 2%); however, the slight influence of the crystallinity percentage in our samples was normalised to 0% using the average of the normalised trends in Figure 9 and Equation (2):(2)$$E\left(0\right)=E\left(x\right)-\frac{E\left(x\right)}{\left(104+x\right)}x$$
where E(0) and E(x) are the elastic modulus at 0% and *x*% crystallinity, respectively, where *x* is the crystallinity percentage.

Once the influence of the PEG, crystallinity, and porosity were normalised in all the samples, the corrected elastic modulus was compared with the theoretical elastic modulus. It was observed that the Mg provides part of its mechanical properties to the matrix, Figure 10.

The results plotted in Figure 10 were used to calculate the bond strength for each composition using the modified Voigt model, Equation (1). The calculated bond strengths are shown in Figure 11. We obtained a similar trend to Cifuentes et al. [27]. Values for the bond strengths were also calculated for the elastic modulus published by Ferrández-Montero [23] (measured with a DMA on tapes) and Pascual-González [22] (measured in a tensile test on 3D printed samples), as shown in Figure 11.

Note that the calculated bond strength is not between the PLA and the Mg, as the interface is PLA-PEI-Mg. The model considers the PLA-PEI-Mg interface as only one interface, but, as shown in Figure 1, there are two interfaces: the PLA-PEI and the Mg-PEI interfaces. To determine which of the two bond strengths has been calculated, we need to refer to the chemistry of the formed bonds. Because the adhesion between the PEI and the PLA is excellent, based on covalent bonds, and because there is evidence of it on the FTIR, it can be determined that the calculated minimum value of the bond strength is between PEI and Mg. 

Regarding SEM images of the fracture surfaces after the tensile tests, Figure 12 shows that the size of the Mg particles is in keeping with the expected values—30 microns—regarding the Mg particle volumetric distribution [26]. Figure 12 shows how some Mg particles remain attached to the compound fracture and some Mg particles detached from the PLA matrix after the tensile tests. This mechanical test produces the worst situation for the interface by significantly enhancing the debonding mechanism between the Mg particles and the PLA. It is well known that stresses on the matrix–particle interface can be up to three times the remote stress—i.e., a stress concentration factor of three [50,51,52]. A fractography of a particle-filled composite broken during a tensile test will likely show the particle fillers detached. It is highlighted that, even after a tensile test where all the particles should detach from the PLA, some particles still remain in the polymer fracture. This result could support a relatively strong PLA-PEI-Mg interface.

From Figure 12, it can also be observed that Mg particles were well dispersed and that no agglomerates were created, even after the FFF processing; this is consistent with the effect of the PEI and the colloidal processing [23,24]. A method for determining when the particles are detached from the matrix was proposed by Lauke et al. The method consisted of studying the deviation from linearity of the stress–strain relation, indicating the loss of reinforcement of the particles to the matrix due to the rupture of the particle–matrix, the interface-debonding mechanism [50,51]. However, in the PLA and PLA-Mg samples studied, the viscoelastic behaviour of the PLA matrix makes it impossible to come to any clear conclusion by analysing the deviations in linearity of the stress–strain graphs.

Once the elastic modulus was understood and the PLA-Mg bond strength was adequately discussed, the tensile strength and fracture toughness were studied. These two properties are highly interrelated, as tensile strength depends on both the fracture toughness and the defects in the material, as expressed in Equation (3):(3)$$K_{IC}=\sigma ϒ\sqrt{\pi a_{c}}$$
where $K_{IC}$ is the fracture toughness, σ is the remote stress, ϒ is a geometrical factor, and $a_{c}$ is the critical crack length. Note that the more significant the crack length or the size of the defect, the lower the required stress for breaking the material; this is known as the tensile strength.

Filaments with a notch were tested to calculate the fracture toughness, *K_IC_*, as shown in Figure 13a,b. The notch tip radius was 1 μm, as shown in Figure 13c. Figure 13d–f provide details of the notched filaments. Interestingly, in Figure 14, an Mg particle was split by the blade. The Mg particle is observed to be well attached to the PLA-matrix; however, this is just for the side on which the blade produced compression during the notching. For the side on which tensile stresses pulled the matrix out due to the cutting movement of the blade, it is observed that the PLA-matrix was wholly separated from the Mg particle.

Regarding the PLA-Mg composites, it is observed that increasing the content of Mg reduces the fracture toughness, as shown in Figure 14. If extrapolated to neat PLA (0% Mg), this indicates that neat PLA should have a higher fracture toughness, and indeed, it does. Comparing neat PLA with PLA-2.5Mg, a considerable decrease in fracture toughness is observed. For a filler to improve the fracture toughness of a material, it must prevent the progress of cracks in the matrix [2]. However, as shown in Figure 14, it was found in all the micrographies that the Mg particles were detached and the PLA-Mg interfaces broken; thus, rather than hampering the development of any cracks, Mg particles could behave as initiators of relatively big cracks (with an average Mg particle size of 30 microns). This would explain the abrupt decrease in the fracture toughness with the addition of just 2.5 wt.% of Mg compared with neat PLA, as there are more significant cracks and easy propagation paths related to the detached PLA-Mg interfaces. It is also observed that the fracture toughness value remains relatively stable—slightly decreasing—with further additions of Mg to the PLA-Mg composites. From another perspective, unlike composites reinforced with fibres, particle fillers do not have large contact surfaces at both sides of a growing crack such that the particles are able to keep the propagating crack closed; in all likelihood, particles will suffer the so-called pull-out failure [53,54].

It was observed that the tensile strength of PLA also decreased with the addition of Mg particles, as shown in Figure 15. Different models were proposed to calculate the theoretical tensile strengths using the values calculated for our PLA-Mg composites. This allowed us to understand the effect of different variables on tensile strength: porosity, PEG, and stress concentration.

The tensile strength was calculated using the same tensile tests to obtain the elastic modulus, as shown in Figure 15.

Before modelling the tensile strength, the effect of different variables was decoupled. Crystallinity was not included as a variable in any model, as any impact on the tensile strength of differences in the crystallinity or types of crystalline forms at room temperature can be disregarded due to the low crystallinity values of our materials; the same reasoning was followed for the elastic modulus. Additionally, any effect that the Mg particles had in improving the tensile strength was discarded because the Mg might behave more as a nucleator of cracks than as an obstacle for the cracks, as discussed in the analysis of the fracture toughness results.

The first proposed model (model 1) considers the possibility that the porosity decreases the effective area of the PLA cross-section. Previously, it was observed that the porosity increased with the addition of Mg. Thus, the volume occupied by the Mg was simplified to behave as pores not supporting mechanical loads. All the pores and Mg-like pores were homogeneously distributed with an infinitesimal diameter size. Using the neat PLA as the reference material, the rest of the theoretical tensile strength values were calculated and are shown in Figure 15. This model does not fit the experimental data, so the tensile strength reduction must be explained by addressing other effects.

A second model (model 2) was analysed, in which the different fracture toughness values calculated for each PLA-Mg composition, as shown in Figure 13, were also considered (together with the increase in the porosity and considering the volume of Mg), as was porosity in model 1. As the tensile strengths are calculated using the measured values of the fracture toughness for each composition, the effect of the PEG is already included in the models through the fracture toughness. Again, using the neat PLA as the reference material, the rest of the theoretical values of the tensile strength were calculated, and are shown in Figure 15. Compared with the experimental values, lower theoretical tensile strengths for the PLA-Mg composites were obtained. This is inconsistent with the theory as, in this second model, the decrease in the tensile strength was underestimated—no consideration of the concentration of stresses around the Mg and pores was included in the model. This second model assumed that the fracture toughness of the PLA matrix remained constant after the PLA-Mg composite was processed. However, it was observed that this second model provided inconsistent theoretical values, indicating that the PLA can be used as a reference material only if more variables are included in the models. Interestingly, to compensate for the low theoretical values obtained in model 2, the fracture toughness of the PLA matrix on the PLA-Mg materials should be higher than that calculated for the neat PLA (0% Mg). This expected increase in the fracture toughness of the PLA matrix is reasonable, considering that the PLA matrix in the PLA-Mg composite has a 5 % weight of PEG, a common effect of plasticisers on polymers [55].

The third and fourth models (models 3 and 4) were proposed. They are similar to models 1 and 2, respectively, but PLA-2.5Mg was used as the reference material for calculating the theoretical tensile strengths for the rest of the PLA-Mg compositions. This is because, in models 3 and 4, it is more realistic to assume that the fracture toughness of the PLA matrix is the same for all the PLA-Mg composites. The calculated results for the tensile strengths are shown in Figure 15.

In the third model (model 3), it is shown again that solely considering the porosity, including the Mg as a pore, is not sufficient to explain the experimental results.

In the fourth model (model 4), a better fit with the experimental results is obtained, and all the results are higher than the experimental ones; this is more consistent with the theory, as model 4 does not consider the stress concentration around Mg particles and assumes the same a_c_ for all the compositions, which are the main differences between the tensile tests with and without a notch. It also indicates that further research is required to properly understand the PLA matrix, as moving from model 2 to model 4 improves the model, because the PLA properties of the PLA matrix are the difference between both models.

Special consideration must be given because, although similar trends on the effect of Mg on the PLA are described by other authors, a direct and reliable trend is hardly possible, compared with other works on PLA. Works on PLA and PLA composites usually do not provide information on the ageing times after the materials were processed [22,56], a variable that can double the tensile strength; it can also take up to three months for the PLA to attain stable mechanical properties, according to our previous work [32]. The same is true for the elastic modulus, glass transition, and almost all of the properties that depend on the amorphous phase. However, the norm “Plastics –Determination of tensile properties– Part 1: General principles (ISO 527-1:2019)” indicates a minimum conditioning time of 16 h before testing, well below the three months required for some commercial PLA studied in the literature; this represents a significant obstacle for a making a comparison in absolute terms between works.

## 4. Conclusions

This paper contributes to a deeper understanding of the effects of Mg particles on PLA-Mg filaments produced through a colloidal route and printed as 1D filaments. The study of the mechanical properties, combined with the DSC, FTIR, and Archimedes analyses, supports the existence of different mechanisms related to the Mg and the PEG, and we understand the effects of those mechanisms on the material’s thermal, chemical, physical, and mechanical properties. This last factor is essential for decoupling the effects of the 3D structure from the effects on the material.

The DSC results for PLA-Mg composites present a complete analysis of the DSC diagrams, extending the discussion of similar DSC graphs in the literature. The DSC analysis offered the following conclusions:-Mg particles seem to enhance the crystallisation of both the α′ and α crystals in the PLA. The FTIR studies concluded this result.-A slight effect on the bands related to the α crystals is observed; however, almost no crystallinity is maintained at room temperature, as expected from the high d-lactide content of the PLA.-The chemical interaction between the PLA and the PEI is consistent with previous work [23] and is essential for the mechanical results.-The stability of -OH-related bands shows the interaction between PLA-PEI.

Regarding the Archimedes tests, the tensile tests, and the fractography, the conclusions are:-A slight increase in the porosity and density results from the addition of Mg.-A good dispersion of the Mg particles through the colloidal route was observed in the SEM fractographies.-The modelling of the Mg particles as a pore provided a good correlation between the theoretical and experimental values for the tensile strength of the PLA-Mg composites. However, it also suggests that the fracture toughness of the PLA matrix in the PLA-Mg composite should be higher than the fracture toughness measured on the neat PLA reference material.

The discussion of the effect of the Mg on the PLA-Mg compositions from this mechanical point of view offers novel insights that have not been described before. This work proposes alternative characterisation methods, and demonstrates them, reaching similar conclusions to other authors. Furthermore, the properties provided for the fracture toughness of PLA-Mg composites have not been published yet.

Without considering whether the materials tested in the literature have been entirely aged or not, similar trends (a comparison in relative terms) for the evolution of PLA-Mg composites can be found [22,23,24,27].

Further research is required regarding the decrease in the fracture toughness and the tensile strength of PLA-Mg composites. Complex models or direct molecular weight measurements are required to correctly calculate the PLA-Mg bond strength.

## Figures and Tables

**Figure 1 polymers-14-05414-f001:**
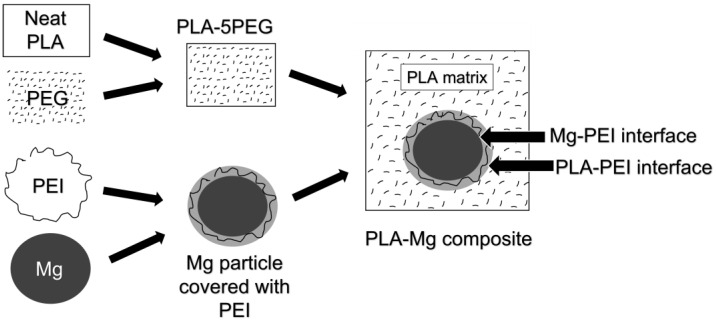
Scheme of the PLA-Mg material produced.

**Figure 2 polymers-14-05414-f002:**
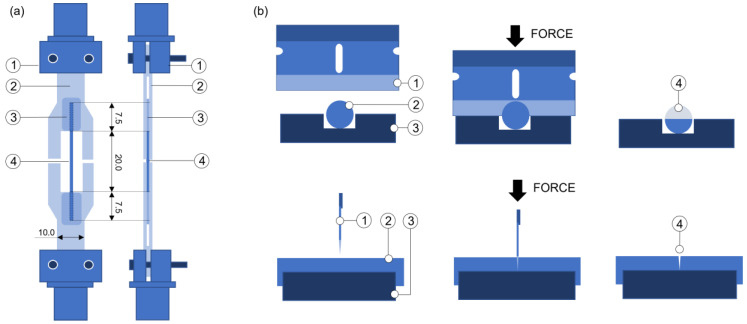
(**a**): Tensile test scheme: (1) mechanical clamp, (2) cardboard, (3) cyanoacrylate-based adhesive as a chemical clamp, (4) PLA or PLA-Mg filament. Units in millimetres; (**b**) notch-making procedure: (1) blade, (2) PLA or PLA-Mg filament, (3) carved steel plate, (4) produced notch.

**Figure 3 polymers-14-05414-f003:**
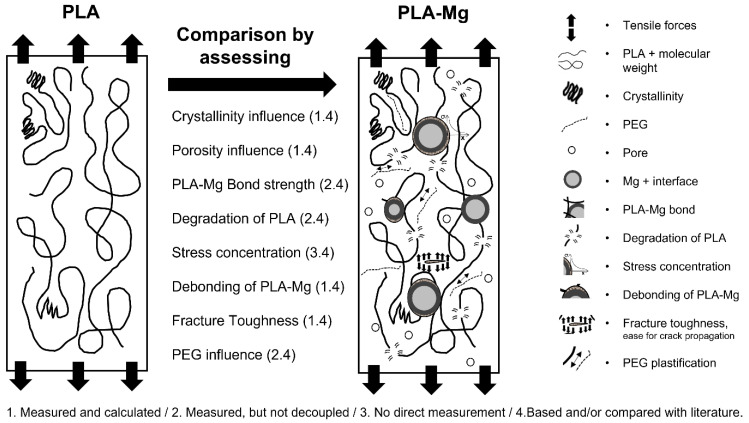
Mechanisms that were considered when determining the tensile strength of the PLA-Mg composites compared with the neat PLA.

**Figure 4 polymers-14-05414-f004:**
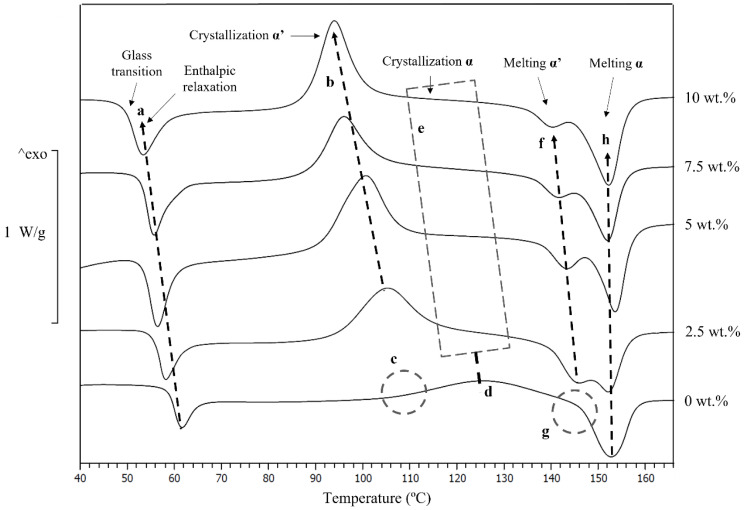
DSC at 10 °C/min of neat PLA and PLA-Mg with 2.5, 5, 7.5, and 10 Mg. (a) Evolution of glass transition and enthalpic relaxation toward lower temperatures, (b) appearance of first crystallisation, shifting towards lower temperatures with the increase in Mg content, (c) missing second crystallisation in neat PLA, (d) crystallisation on PLA without Mg, (e) uncertain shape of crystallisation on PLA-Mg composites, (f) first melting appeared in samples with Mg, (g) missing a melting peak in neat PLA, (h) melting point variation with Mg content.

**Figure 5 polymers-14-05414-f005:**
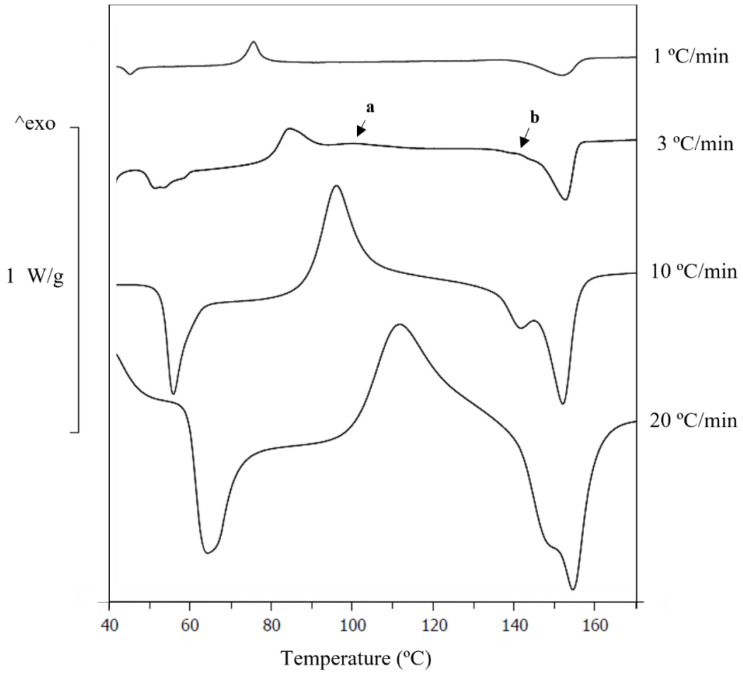
DSC of PLA-7.5Mg at different heating rates from 1 to 20 °C/min. Scale bar for material heated at 10 °C/min. For PLA-7.5Mg at 3 °C/min: (a) α crystallisation, (b) transformation of α′ into α at the same time α′ is melting.

**Figure 6 polymers-14-05414-f006:**
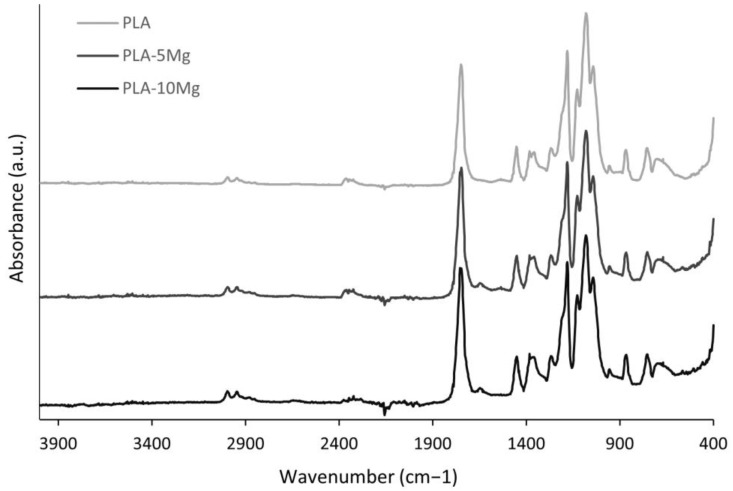
FTIR spectrum of PLA, PLA-5Mg, and PLA-10Mg.

**Figure 7 polymers-14-05414-f007:**
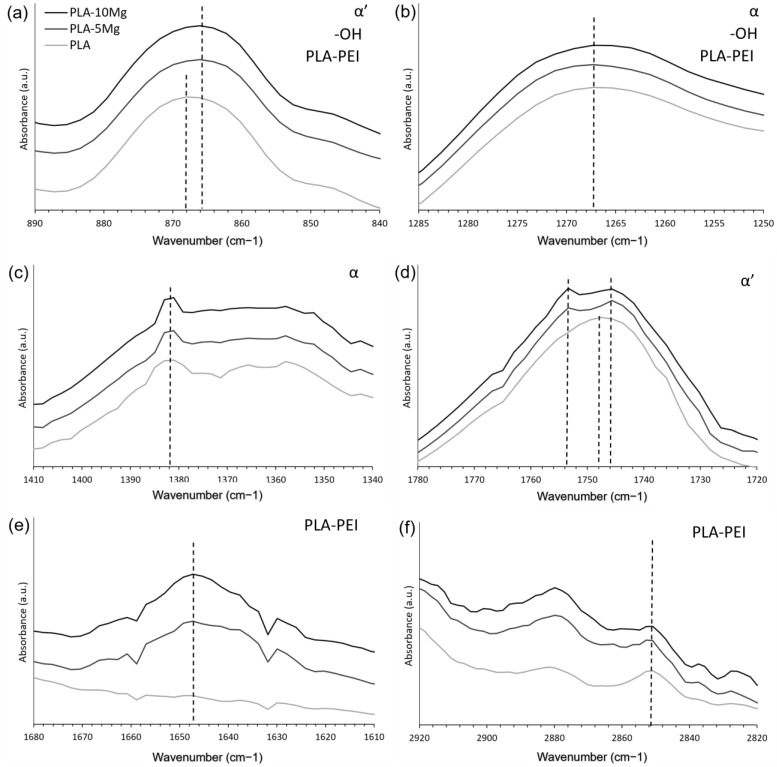
Details of FTIR of neat PLA and PLA-Mg composites, used to study α and α′ forms related peaks, -OH terminal groups, and PLA-PEI interactions. The studied effects related to that peak are shown in the upper right corner of each FTIR. (**a**) 868 and 866 cm^−1^, (**b**) 1266 cm^−1^, (**c**) 1382 cm^−1^, (**d**) 1746, 1748 and 1754 cm^−1^, (**e**) 1647 cm^−1^ and (**f**) 2850 cm^−1^.

**Figure 8 polymers-14-05414-f008:**
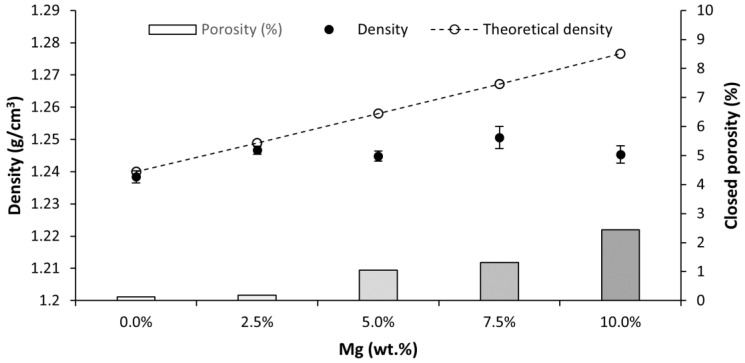
Density and closed porosity compared with the weight percentage of Mg content in the PLA and PLA-Mg composites.

**Figure 9 polymers-14-05414-f009:**
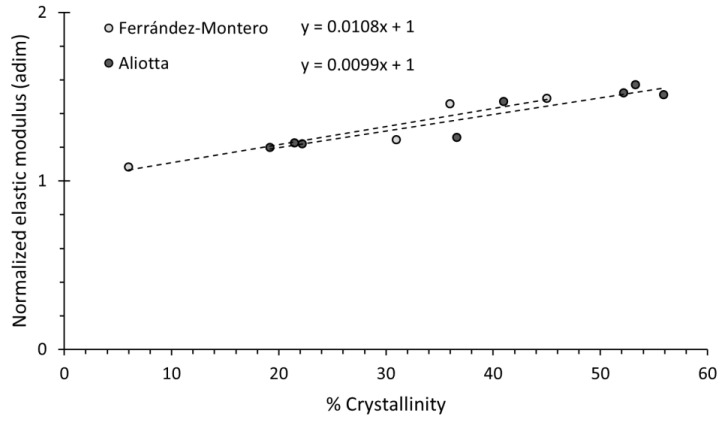
Normalised values of the elastic modulus for PLA materials and their evolution with the crystallinity percentage.

**Figure 10 polymers-14-05414-f010:**
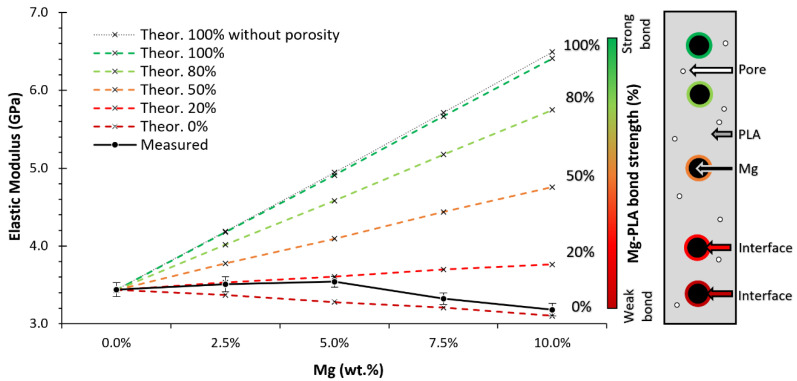
Elastic modulus versus the Mg content and its theoretical value, considering the porosity and the PLA-Mg bond strength. A bond strength of 100% refers to a perfect interface in which all loads are transmitted to the Mg particle; 0% represents a weak interface where loads are not transferred from the PLA-matrix to the Mg particle.

**Figure 11 polymers-14-05414-f011:**
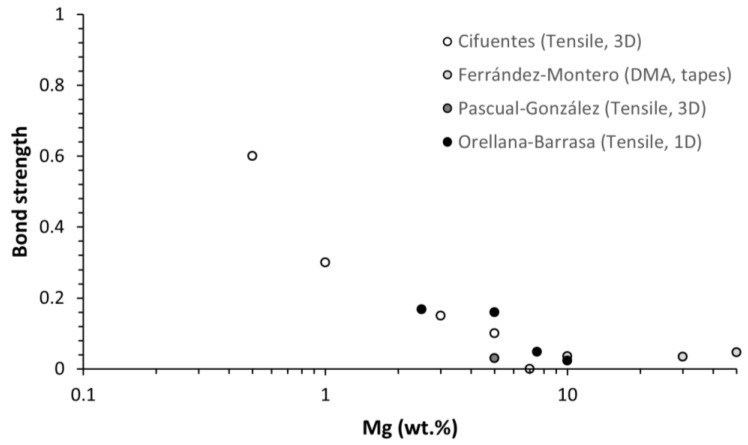
Lower limit values of PLA-Mg bond strengths calculated from the Voigt model. Data derived from Cifuentes [27], Ferrández-Montero [23], Pascual-González [22], and Orellana-Barrasa.

**Figure 12 polymers-14-05414-f012:**
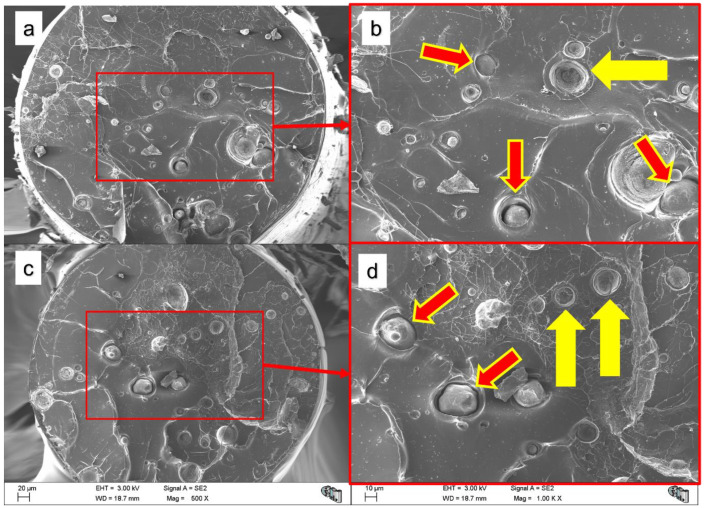
SEM fractographies of PLA-5Mg after tensile tests at 1 mm/min. (**a**) PLA-Mg tensile test fracture surface, (**b**) detail of fracture surface. (**c**) Another PLA-Mg tensile test fracture surface and (**d**) a detail of it. Arrows indicate that the Mg particles detached from the PLA-matrix after a tensile test.

**Figure 13 polymers-14-05414-f013:**
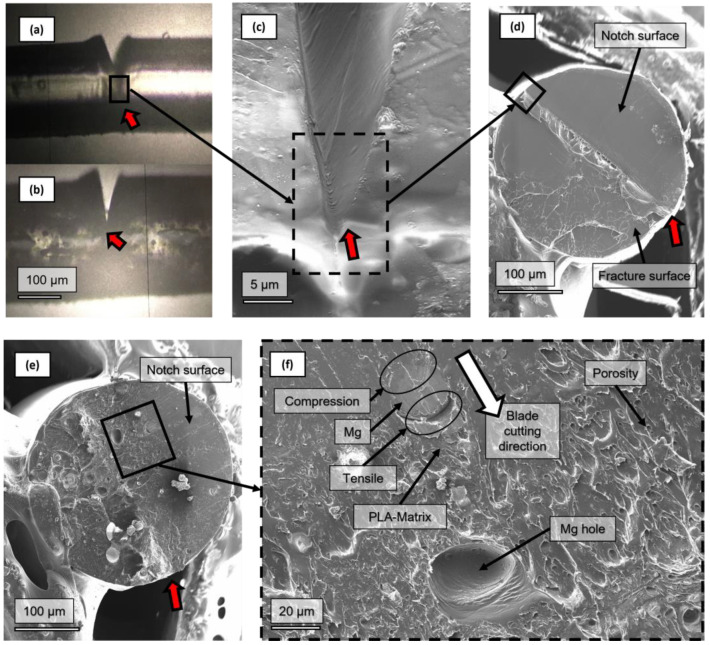
Detail of the notch made on the PLA and PLA-Mg samples and the fracture surface: (**a**) PLA filament with notch; (**b**) PLA-Mg filament with notch; (**c**) detail of the notch from a lateral view and a notch tip radius of 1 μm; (**d**,**e**) notch and crack surface after tensile tests of PLA and PLA-5Mg, respectively; (**f**) detail of the PLA-5Mg notch surface. Small arrows indicate the edge of the notch.

**Figure 14 polymers-14-05414-f014:**
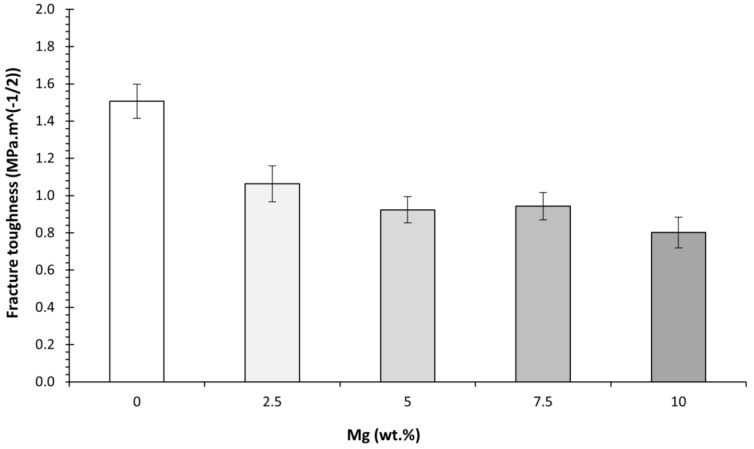
Evolution of the fracture toughness of PLA-Mg composites with Mg content.

**Figure 15 polymers-14-05414-f015:**
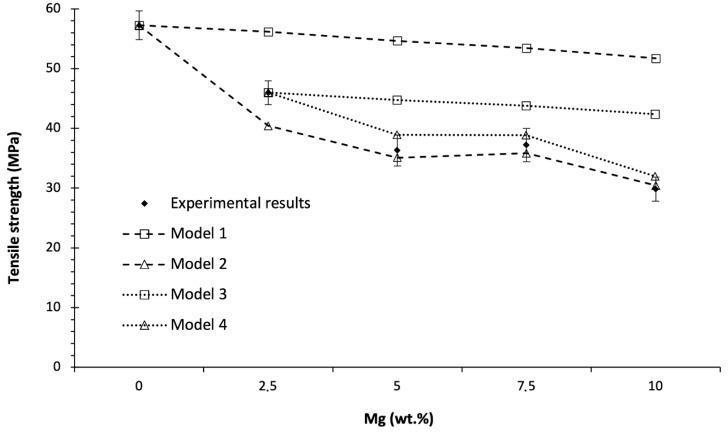
Evolution of the tensile strength of the PLA-Mg composites with the content of Mg particles. Theoretical values of the tensile strength are provided regarding different models. Experimental results are shown in bold.

**Table 1 polymers-14-05414-t001:** Thermal properties of extruded 1D PLA and PLA-Mg filaments at a scanning rate of 10 °C/min. T_g_: glass transition; T_ER_: enthalpic relaxation temperature; ΔH_ER_: enthalpy of enthalpic relaxation; T_CC.1_ and T_CC.**2**_: cold crystallisation temperature of two crystalline phases, respectively; T_m.1_ and T_m.**2**_: melting temperature of α′ and α crystals, respectively. Following the IUPAC convention, positive enthalpy changes indicate that the material absorbs the energy, which indicates an endothermal process. Melting and crystallisation enthalpies are not included in the table, as providing an accurate value was tricky due to the overlapping crystallinity values below 2% for all the samples.

%Mg	Heating Rate (°C/min)	T_g_ (°C)	T_ER_ (°C)	ΔH_ER_ (J/g)	T_CC.α′_ (°C)	T_CC.α_ (°C)	T_m.α′_ (°C)	T_m.α_ (°C)
0	10	59.9 ± 0.5	62.6 ± 0.5	4.5 ± 0.1	-	125 ± 1	-	151 ± 1
2.5	10	54.9 ± 0.5	58.0 ± 0.5	5.1 ± 0.1	105 ± 1	-	146 ± 1	152 ± 1
5.0	10	52.3 ± 0.5	56.4 ± 0.5	8.6 ± 0.2	100 ± 1	-	143 ± 1	153 ± 1
7.5	10	49.8 ± 0.5	53.3 ± 0.5	7.0 ± 0.2	96 ± 1	-	142 ± 1	152 ± 1
10	10	48.7 ± 0.5	53.3 ± 0.5	8.1 ± 0.2	94 ± 1	-	140 ± 1	152 ± 1

**Table 2 polymers-14-05414-t002:** Thermal properties of the extruded 1D PLA and PLA-Mg filaments. T_g_: glass transition; T_ER_: enthalpic relaxation temperature; ΔH_ER_: enthalpy of enthalpic relaxation; T_CC.**α**′_ and T_CC.**α**_: cold crystallisation temperature of α′ and α crystals, respectively; T_m.**α**′_ and T_m.**α**_: melting temperature of α′ and α crystals, respectively. Following IUPAC’s convention, positive enthalpy changes say that the material absorbs the energy, which indicates an endothermal process. Crystallinity values are below 2% for all the samples.

%Mg	Heating Rate (°C/min)	T_g_ (°C)	T_ER_ (°C)	ΔH_ER_ (J/g)	T_CC.α′_ (°C)	T_CC.α_ (°C)	T_m.α′_ (°C)	T_m.α_ (°C)
7.5	1	42.8 ± 0.5	45.0 ± 0.5	2.4 ± 0.1	76 ± 1	-	-	152 ± 1
7.5	3	48.2 ± 0.5	50.7 ± 0.5	4.6 ± 0.1	84 ± 1	100 ± 1	-	153 ± 1
7.5	10	49.8 ± 0.5	53.3 ± 0.5	7.0 ± 0.2	96 ± 1	-	142 ± 1	152 ± 1
7.5	20	59.8 ± 0.5	64.2 ± 0.5	8.5 ± 0.2	112 ± 1	-	147 ± 1	154 ± 1

## Data Availability

Not applicable.

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
