# Peer review of "The Mechanical, Thermal, and Chemical Properties of PLA-Mg Filaments Produced via a Colloidal Route for Fused-Filament Fabrication"

_polymers, 2022, doi:10.3390/polym14245414_

Round 1
Reviewer 1 Report
The manuscript investigated the thermal, chemical, and mechanical properties of PLA composites with the incorporation of Mg particles, which were made into filaments expected for additive manufacturing. However, as it is described in the abstract of the manuscript, it seems that some tests were just carried out. The novelty of the manuscript looks weak. 1) The idea is not new. Some works on PLA/Mg filaments for 3D printing have been published. See examples: Processing and properties of PLA/Mg filaments for 3D printing of scaffolds for biomedical applications, Rapid Prototyping J 28(5) (2022) 884-894. Study of the matrix-filler interface in PLA/Mg composites manufactured by Material Extrusion using a colloidal feedstock, Additive Manufacturing 33 (2020). Magnesium Filled Polylactic Acid (PLA) Material for Filament Based 3D Printing, Materials 12(5) (2019). 2) The contribution from the manuscript is not impressive. a) The goal of coating Mg particles with polyethyleneimine in the manuscript seems to enhance the bonding between Mg and PLA, which could be one of highlights of the manuscript. However, it was proved that such bonding was quite weak, and the mechanical properties of composites were deteriorated. In comparison, the mechanical properties of PLA/Mg composites in the literature were almost not changed (Processing and properties of PLA/Mg filaments for 3D printing of scaffolds for biomedical applications, Rapid Prototyping J 28(5) (2022) 884-894.) b) The thermal and chemical properties of PLA‐Mg filaments have been reported in the literature.
Author Response
- "The novelty of the manuscript looks weak. The idea is not new."
We are aware that other authors have previously worked on PLA-Mg composites. We repeat part of other authors' works but study 1D filaments. This lets to decouple the variables that affect the 3D structure from those solely related to the material. We confirm, support and extent the existing literature with novel results like the measured fracture toughness on the filaments.
- "The contribution of the manuscript is not impressive."
This work supports and extends the existing literature on PLA/Mg composites, advancing the knowledge of this material and clarifying the required future research after a detailed analysis of the literature from this new perspective.
2.a.1. "However, it was proved that such bonding was quite weak, and the mechanical properties of composites were deteriorated."
The calculated bond strength through the Voigt model could be weak, but that is a limit lower as the degradation of the PLA cannot be decoupled. The text has included a detailed explanation, clarifying how to interpretate the calculated bond strength.
2.a.2. "In comparison, the mechanical properties of PLA/Mg composites in the literature were almost not changed."
It has included a comparison with the mechanical properties reported in the literature.
2.b. The thermal and mechanical properties of PLA-Mg filaments have been reported in the literature.
But have not been discussed as much as we do; thus, we are setting the knowledge of what it is said "this could be…" to something more like "this is this". Moreover, 1D filaments have not been studied, but 3D structures. We are decoupling the effect of the structure on the material, which is necessary for a better understanding of 3D-printed structures. Fracture toughness, modeling of PLA/Mg tensile test, and tensile test fractographies have never been reported.
Thanks to your feedback, I am grateful as the paper's quality has improved. Thank you for all the questions, comments, and suggestions.
Reviewer 2 Report
-
1.The experimental data are well analyzed, but the material preparation and experimental process are less described. In addition, the research lacks application prospects, such as biocompatibility in the background.
- 2.The effect of Mg addition on print quality was not characterized.
- 3.For additive manufacturing, the performance of forming parts is more important, so the paper needs to increase the characterization of material forming parts.
Author Response
We have updated the figures to provide further information. Regarding material preparation, there is a reference providing the detailed composition of the material, and we are not allowed to include more information in this document.
2 and 3.
This study focuses on understanding the properties of PLA/Mg materials. For this reason, just 1D filaments have been studied for decoupling the effects solely related to the material from those related to the 3D structure. Therefore, a study of the printing quality of 3D structures has not been done. This work will support future studies of how different 3D printed structures behave, and it will be easier to assess the impact of the parameters related to the 3D printing process.
Thank you for all the questions, comments, and suggestions.
Reviewer 3 Report
The paper is about PLA/Mg composites for additive manufacturing for Healthcare applications. The paper shows a significant amount of work with adequate characterizations. However, the results do not show the interest of using Mg in PLA (no experience on the application part and the results are not encouraging from a mechanical point of view).
Some points of improvement:
1) Explain why to use the colloid route.
2) the paragraph on characterizations in the introduction is not very useful: to be re-diluted in the Materials & Methods part.
3) the material production part needs some details. Currently, it is not possible to reproduce the method with the given information.
4) Why use PEI? Knowing that this seems to be the weak point of the mixture cohesion.
5) Add PEG in the scheme (figure 1).
6) Try tests without PEG to determine the Mg impact and the PEG impact, especially on thermal properties and crystallinity.
7) Demonstrate that the results are significant in FTIR tests (reproducibility and repeatability).
8) In the end, no real test from the FFF process.
Author Response
- Explain why to use the colloidal route.
The colloidal route ensures a perfect dispersion of the Mg particles inside the PLA as Mg particles are coated with PEI, and PLA is precipitated around every individual PEI-Mg particle. This ensures that no Mg agglomerates are formed.
- The paragraph on characterization in the introduction is not very useful to be re-diluted in the Materials & Methods part.
Ok, thank you, we have worked out that part and merged it into the methods.
- The materials production part needs some details. Currently it is not possible to reproduce the method with the given information.
We have included as much information as we have been allowed. References with more details about the materials and production have been specified.
- Why to use PEI?
A clarification of why to use PEI has been included in the text.
Different options are described in the literature to improve the PLA-Mg adhesion (CTAB, vitamin E, PEI…). PEI was found to provide the best PLA-Mg bond in the literature. As described in the literature, PEI also ensures an excellent dispersion of the Mg particles inside the PLA as it creates a voluminous and charged layer around the Mg during the colloidal processing, ensuring that no agglomerates are created. It was proved in this work that it also avoids the formation of Mg agglomerates after the FFF process.
5) Add PEG in scheme in figure 1.
Figure 1 has been removed to avoid confusion, as the production of the material is provided in great detail in the cited literature.
6) Try tests without PEG to determine the Mg impact and the PEG impact, especially on thermal properties and crystallinity.
Undoubtedly, studying the PLA-PEG material will help to decouple the effects of PEG and Mg on thermal properties and crystallinity. In this work, as the content in PEG is constant in all 4 PLA-Mg materials (5 wt.% respect the PLA weight), it can be indirectly decoupled. It has been compared with the work of Pascual-González et al. as a reference to understand the effect of PEG.
7) Demonstrate repeatability of FTIR results.
The FTIR results were coherent with the existing literature. We publish them as they support our work, especially when looking for the -OH terminals. The fact that it is coherent with all the existing literature provides security that the FTIR results are reliable. We performed different runs and changed variables like the number of scans and the resolution… We are providing the most detailed scans coherently with the rest of the runs and repeatable.
8) In the end, no real test from the FFF process.
The corrected version of the article tries to avoid misunderstanding the objective of the research.
The research focused on the PLA/Mg, which has been FFF processed to produce 1D filaments. The PLA/Mg was processed following the indications provided by COLFEED (155 ºC). No influence of the FFF parameters has been studied (printing temperature, nozzle diameter…), as single 1D filaments were produced. Producing 1D structure leads to a clearer understanding of the properties of the printed PLA/Mg material decoupled from the 3D structure (layer height, printing direction, printing speed, infill, …). Our main goal is to understand the PLA/Mg after the FFF process, which will later help to understand the 3D-printed structure.
Thanks to your feedback, I am grateful as the paper's quality has improved. Thank you for all the questions, comments, and suggestions.
Reviewer 4 Report
The work is very interesting and concerns the stages of PLA composites with mineral fillers. The proposed idea for making the composite is interesting. The authors showed creativity and good technique of the researcher. It is worth adding here that similar works have already been published on this topic, but it does not reduce the value of the work. The literature review is improving and provides a good background for the goal set. It is true that the filling is a maximum of 10% in mass. This has managed to observe an interesting phenomenon. The drawings and descriptions are correct and the final conclusions confirm the obtained results. Generally, the work should be prepared in accordance with the requirements of the editorial office.
Author Response
Thank you so much for your comforting words.
Round 2
Reviewer 1 Report
The manuscript looks fine.
Author Response
English language and style have been improved by the revision of a native expert.
Reviewer 3 Report
Please send a finalized version of the document (with visible changes and comments if any, in English).
Author Response

(The authors gave the same response as above.)
